# Assessment of perceived patient comfort and ease of bedpan handling by caregivers, a cross-sectional survey

**Pia Secher Cailleteau, Lucie Cadon, Cécile Paille, Elise Olivier, Thomas Rulleau** *

CHU Nantes, Movement - Interactions - Performance, MIP, UR 4334, Nantes Université, Nantes, France

* thomas.rulleau@chu-nantes.fr

## Abstract

### Introduction

Since its creation in the 18th century, bedpan has remained more or less the same. Its unique material composition varies from one model to another, but its shape remains relatively similar. The environment is one of the four pillars of the nursing paradigm. It is therefore essential to question this device in the nursing discipline.

### Aim

To assess perceived patient comfort and ease of bedpan handling by nurses and their assistants.

### Method

A cross-sectional survey via an online questionnaire was conducted among nurses and nursing assistants, nursing students, and health executives using the bedpan for their patients to assess their feelings and their level of satisfaction. The questionnaire asks professionals about the ease of handling the bedpan and the patient's perceived comfort.

### Results

431 responses were obtained out of 3007 persons interviewed (14.3%). 83.0% believe that the cause of poor elimination by the patient is often or very often due to physical discomfort on the bedpan. 62.6% find the installation of the bedpan rather tough or very difficult. 59.2% find the removal of the bedpan rather tough or very difficult.

### Discussion

Our study confirms our hypothesis and highlights a lack of comfort in the bedpan as perceived by professionals. This analysis is the first step in enabling the nurse researcher to support change in the transformation paradigm.

**Data Availability Statement:** All relevant data are within the paper and its Supporting information files.

**Funding:** The authors received no specific funding for this work.

**Competing interests:** The authors have declared that no competing interests exist.

## Introduction

In health establishments, hygiene care represents a large part of nursing care in general [1]. In France, It is carried out by nursing assistants and nurses, as part of their role, according to the French Public Health Code [2, 3]. This care considers the elimination of the patient's stools and urine. When a patient with an acute loss of autonomy cannot get up, it is necessary to use a bedpan. In France, the most common bedpan is made of a single material: polypropylene, but there is also a metal one. In both cases, it has an oblong shape with a seat in its narrow part, an opening for the collection of urine and stool in its widest part, and a handle at the widest end. It is called "bedpan" but corresponds to a « fracture bedpan » in opposition to the regular one in English. Since the 18th century, it has remained more or less the same [1, 4]. Various points of interest were studied with patients on the bedpans, such as stress, practical usage, comfort, or pain. Nursing research and teaching have evolved significantly in recent years [5, 6] In this context, nurses must play their full part in questioning practices with the accompanying theoretical support [7].

A British team interviewing orthopedic patients pointed out that, firstly, the bedpan is a cause of stress; secondly, it is not big enough and « squeezes their bums »; and thirdly, the metal bedpan is noisier than the plastic one [8]. A German-Swiss study [9] on the practical use of the bedpan found that patients bypassed the bed restriction and endangered themselves by going to the bathroom to avoid using the bedpan. Others, unable to use the bedpan, could experience acute urinary retention or suffer from constipation, requiring additional care (use of a urinary catheter, administration of laxative treatments). The position and discomfort on the bedpan can also lead to the retention of urine or stool and thus cause behavioral problems in the elderly, confusion, or pain. It alters the local skin condition, particularly at the pressure points (sacrum, coccyx, seat bones). Stage 0 or 1 pressure sores are common when bedpans are used, as the rigid surface of the bedpan impairs blood circulation and thus local tissue oxygenation. When the bedpan is removed, redness may be visible, which does not disappear immediately [10]. The people mainly concerned are the elderly, subjects whose body mass index (BMI) is less than 18, undernourished subjects, or even polypathological subjects. The International Association for the Study of Pain [11] describes pain as an unpleasant sensory and emotional experience associated with actual or potential tissue damage or described in terms of such damage. The pain would be a consequence of the decubitus dorsal and the rigid material that constitutes the bedpan, leading to the consumption of analgesic treatments [9].

The bedpan is problematic at different points: urinary elimination needs to be completed, it causes pain, and it is uncomfortable to use by the patient and caregiver. Furthermore, setting up the bedpan takes work. The caregiver must repeat this several times. The nursing activity causes pain to the caregiver and can degenerate into musculoskeletal disorders [12, 13].

According to The International Association for the Study of Pain, "pain is the greatest enemy of comfort" [11]. The concept of comfort since F. Nightingale in 1863 [14] emphasises the importance of global health care as a multidimensional concept. It is not only physical and physiological but also psychological and environmental [15]. The bedpan creates many discomforts and pain issues for patients and caregivers but also stress and a sense of dependency for some patients [9]. The bedpan needs to satisfy the concept of comfort in this multidimensional goal. Additionally, according to Virginia Henderson, the basic needs of "eliminating bodily waste" and "avoiding danger" are unmet in these situations [16–18]. In addition to this historic research of comfort, the nursing metaparadigm highlights four concepts: person, environment, health, and nursing [5, 7]. Although the bedpan has been used for a long time, its use does not seem to have been questioned in the light of the nursing metaparadigm, based on the four concepts of the nursing metaparadigm.

To date, several studies evaluate patients' perspectives on bedpans, but no survey of caregivers' perception of the bedpan has been found in the literature. In this exploratory study, our objectives are to investigate, in the light of nursing metaparadigm concepts, the comfort of the patient perceived by the professional and the ease of manipulation of the bedpan by the professional.

## Methods

### Population

The survey population includes professionals used to installing the bedpan, nurses, and caregivers (nursing assistants) of the same University Hospital.

**Inclusion criteria.** Nurses and caregivers work in the medicine and surgery departments and aftercare service of the same University Hospital.

**Exclusion criteria.** Nurses and caregivers work in units where the bedpan is not in use in this University Hospital (i.e., consultation units, psychiatry units, pediatric and neonatology departments, operating theatres, sterilisation departments, and physical medicine and neurological rehabilitation departments).

### Design

We conducted a cross-sectional exploratory, descriptive study using a self-administered online questionnaire to evaluate different items using a Likert scale. The questionnaire ran from January 31, 2022, to March 1, 2022, and was distributed to professionals: managers, nurses, caregivers, and students via professional emails. A QR code with the link to the questionnaire was also displayed in the break rooms of the departments concerned, as well as a direct link on the home page of the University hospital intranet. We targeted the medical and surgical departments and the aftercare service (i.e., 3,007 professionals). The questionnaire was distributed for one month with a reminder by email 15 days after the first sending. The responses are anonymous.

This questionnaire included mutually exclusive closed questions formulated using a Likert scale and open questions. Seven nurses and nurses' assistants received a test questionnaire to check the questions' relevance and intelligibility. This pilot evaluation validated that the questionnaire had been properly understood, without any misinterpretations.

The final questionnaire consists of 18 questions (S1 Appendix), divided into three sections in emphasis with four nurse metaparadigm concepts (person, environment, health, nursing) [7]. The first section emphasis nursing and concerns the profile of the health professionals questioned; the variables studied are the profession (qualification as a nurse or assistant), the age group, the number of years' experience of the caregiver using bedpan, and the frequency of use per week. The second section emphasis environment concerns the assessment of the caregiver's facility to handling the bedpan; the variables studied are the difficulty of installing the bedpan, the problem of removing the bedpan, the comfort of the handle, and the frequency of overturning the bedpan. The last section focuses on the patient and health (especially well-being) as perceived by the professional performing the care, with the following variables: the time the patient spent on the bedpan, the complete elimination or not of the patient at the time of withdrawal of the bedpan, and the possible causes of incomplete elimination.

### Ethics

This study complies with the Helsinki Declaration. French law makes no provision for an ethics committee to be involved in this study. Healthcare professionals gave their written consent to participate by voluntarily completing the questionnaire.

## Statistical analysis

The whole questionnaire was entered on the Sphinx® software version IQ2. The quantitative data were processed using descriptive statistics tools using the statistical software Sphynx® version IQ2. The results were presented in the form of tables of numbers and frequency of the characteristics of the individuals as well as their perception of the bedpan. The respondents' verbatim comments were classified as main ideas using Excel software. We considered a response rate greater than 10% statistically significant with an estimated number of target subjects of 3007 within the abovementioned departments.

## Results

### Participants

Out of 3007 nurses and caregivers concerned, 431 answered our questionnaire (80% via email, 20% via QR code). Four were excluded from the analysis because they only answered the "Profile" part. 51.0% of the respondents are nurses, 45.0% are caregivers, and 3.0% are students in nursing school.

### Descriptive findings of the study

Analysis of the results is structured according to categories developed by the author of the questionnaire: Profile of respondents, Bedpan manipulation, caregivers' appreciation of bedpan handling, patient's well-being: time spent on the bedpan and cause of incomplete elimination, and comments on verbatim.

Table 1, Figs 1–3 present results for all items concerned.

**Profile of respondents.** Women represent 88.4% of respondents, which reflects the care profession. More than half respondents (57.1%) are between 31 and 50 years old. 61.0% of respondents have an experience in bedpan use over ten years. Nurses and their assistants use the bedpan several times daily (49.0%) (Table 1).

**Table 1. Respondents' profile (n = 431).**

| | |
|---|---|
| Men | 11.4% |
| Women | 88.4% |
| Between 18 and 30 yo | 26.5% |
| Between 31 and 50 yo | 57.1% |
| More than 50 yo | 16.5% |
| Respondents' profession | |
| Nurses | 51.0% |
| Caregivers | 45.0% |
| Students | 3% |
| Duration of use of the bedpan | |
| < 1 year | 4.6% |
| Between 1 and 3 years | 9.7% |
| Between 4 and 10 years | 24.6% |
| > 10 years | 61.0% |
| Frequency of use of the bedpan | |
| Several times a day | 49.0% |
| Several times a week | 37.1% |
| Less than once a week | 8.1% |
| Less than once a month | 5.8% |

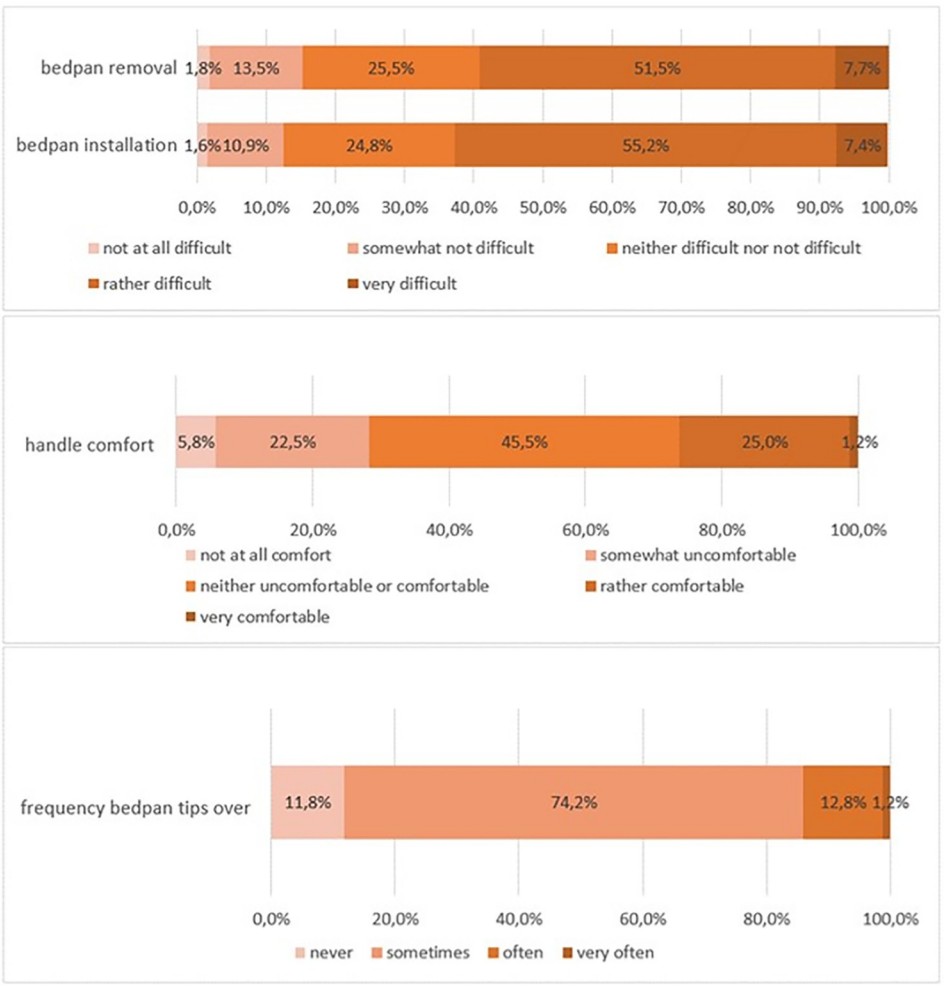

**Fig 1. Caregivers' appreciation of bedpan handling (n = 431).**

**Bedpan manipulation and caregivers' appreciation of bedpan handling.** Installation and removal are rated as "difficult or very difficult" at 62.6% for installation and 59.2% for disposal. The handle is neither comfortable nor uncomfortable for 45.5% of respondents or not at all comfortable and rather not comfortable for 28.3%. Moreover, the bedpan falls over "sometimes" for 74.2% and "often" for 13.0% (Fig 1). It means that respondents admit that urines and stools end up in the bed.

**Patient's well-being: Time spent on the bedpan and cause of incomplete elimination.** More than half of the respondents remove the bedpan after 6 to 10 minutes (57.1%) of use. 11.9% estimate removal after 11 to 15 minutes. On the contrary, 30.5% estimate the removal before 5 minutes.

Elimination is incomplete when the bedpan is removed, according to 51.6% of professionals (Fig 2). 63.1% of them estimate it is never because of lack of time. 83.0% consider that discomfort is "often" or "very often" the reason for incomplete elimination. (Fig 3). The decubitus dorsal is also reported as a cause of pain in comments by 32.0% of professionals.

**Comments on verbatims.** Out of 179 caregivers who left a free comment: some find the bedpan too rough (16.0%), uncomfortable (13.0%), unsuitable for the morphology of patients

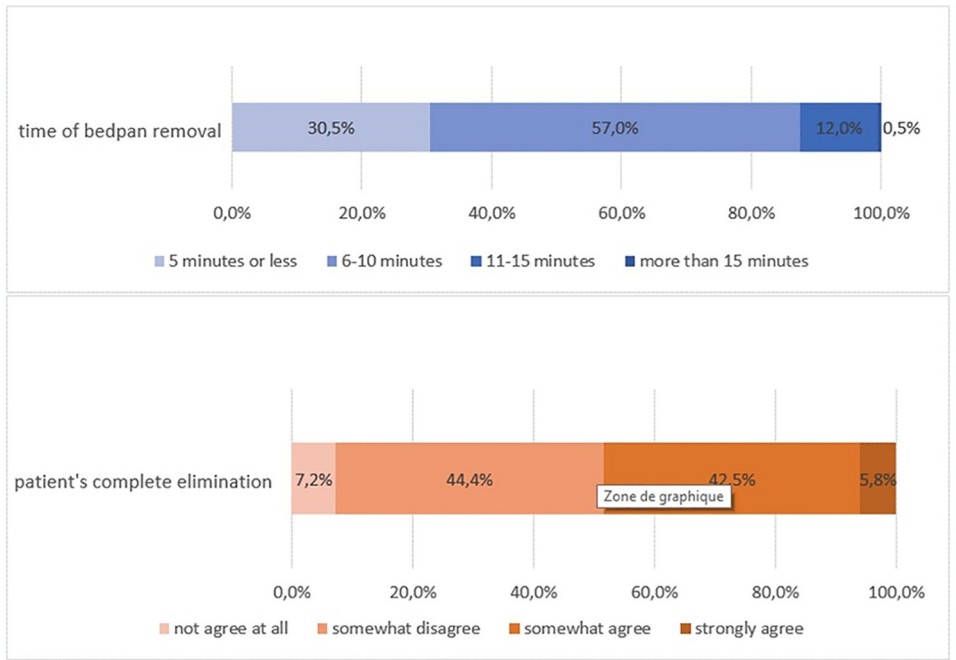

**Fig 2. Patient well-being (n = 429 and n = 428).**

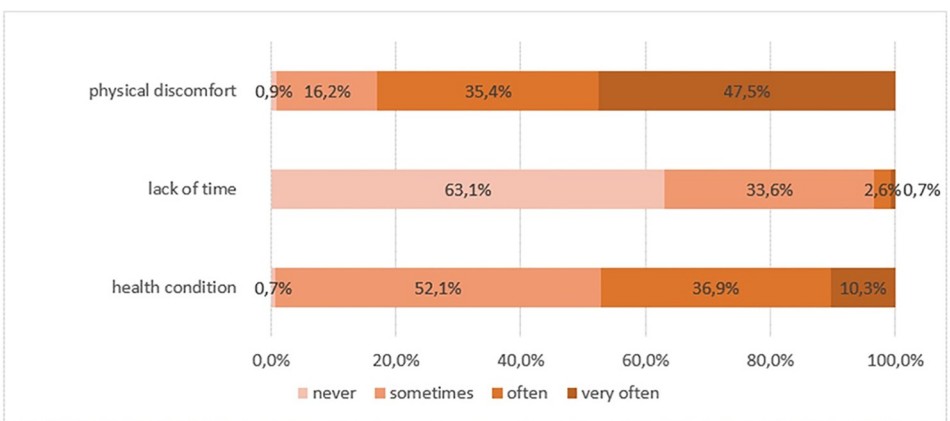

**Fig 3. Cause of incomplete elimination.**

(11.0%), and presents a risk of irritation or bedsores (9.0%). Among the ideas for improvement, out of 104 comments, 24.0% suggested a gel cushion or padded support, 22.0% a more anatomical shape, and 21.0% a more comfortable material.

## Discussion

This study investigated caregivers' perceptions of bedpan use, in terms of the four concepts of the nursing metaparadigm (nursing, environment, health, and patient). This study investigates patient comfort perceived by caregivers and the ease of bedpan handling by caregivers. The

leading information is that 6 out of 10 caregivers have difficulty handling the bedpan during installation and removal. They sometimes have to go over it several times before the patient feels stable on it. When removing the bedpan, care must be taken not to tip it over, or risk introducing urines and stools into the bed. More than 80.0% of the professionals feel that the bedpan's lack of comfort contributes to poor elimination.

More than 14.0% of the people targeted responded to our survey. This significant percentage underlines the caregivers' interest in care quality and the patient's well-being. Usually, in other studies, such as Gattinger et al.'s or Cohen's, the patients were interviewed because they were naturally directly concerned. It is interesting to show how professionals perceive patients' feelings and how close they are [8, 9]. When those surveys studied 87 or 10 patient samples, this study got a sample of 431 professionals. Of 431, 88.4% are women, highlighting that nursing care is still mainly female. 57.1% are between 31 and 50 years old, and 61.0% have an experience in bedpan use over ten years. Nurses and assistants use the bedpan for their patients several times a day. With those three data, we can say that professionals are used to manipulating the bedpan.

In emphasis with environment concept, as for person and caregivers, regarding bedpan manipulation, professionals agreed on the capacity of the bedpan to fall over during use by patients. Gattinger et al. bring up in their survey patients' fear "that everything may go bedside" which means urine and stools, and the fact that "they slide off the bedpan". If this occurs, the "degree of inconvenience" would be "unacceptable" for them [9]. Moreover, in our survey, professionals find the installation and the removal both "difficult" and "very difficult". Some verbatim insist on the non-ergonomic shape of the bedpan and its handling. It could lead to or increase musculoskeletal disorders among caregivers who are already at risk [12, 13].

In emphasis with patient and health, time spent on the bedpan is estimated to be more than five minutes for the two third of respondents. It could be related to the necessity to be sure that the patient eliminates completely, which is not the case according to 51.6% of respondents. On the contrary, patients report in Gattinger et al.'s survey that if they experienced "a long time until bedpan was removed" (10.0%), it would be "unacceptable" for 64.0% of them [9]. This time spent on the bedpan therefore seems to contradict the need for patient well-being in care. The patient's environment becomes a source of aggression that needs to be corrected.

Finally, the patient's well-being seems to be a genuine concern for professionals. Incomplete elimination may result from the installation or the dorsal decubitus position, even if caregivers give time to patients on the bedpan. Indeed, discomfort is always pointed out by caregivers. Most of them (99.3%, 10.3% "very often") agreed that the patient's health condition is one of the causes of incomplete elimination. Still, they agreed (99.1%, 47.5% "very often") that physical discomfort is the lead cause and directly attributable to the bedpan material and design. In *Orthopaedic patients' perceptions of using a bedpan* [8], over ten patients, one told him that a plastic bedpan "squeezes up [its] bum", and one reported that she mentioned the pain to the nurse. Still, she did "not feel able to mention anxiety", she said. Furthermore, in *Patient experience with bedpans in acute care* [9], "the pain is mainly caused by sitting on the bedpan for an extended period of time" (66%). Another cause less mentioned (48%) was "lying in a supine position", and 41% described "pain as a result of their physical constitution". According to Gattinger's study, patients found the bedpan hard (81%) and cold (67%) but also not deep enough (42%), like one of Cohen's patients [8, 9]. In our survey, free comments from 179 caregivers underline the bedpan's negative characteristics: too rough (16.0%), uncomfortable (13.0%), unsuitable for the morphology of patients (11.0%), which presents a risk of irritation or bedsores (9.0%).

We can see that the healthcare professionals surveyed have a perception of patient discomfort that coincides with the responses of the German-Swiss study [9]. Caregivers have full of

resources to try to improve comfort at the use of the bedpan, such as "care oil" and "towel under the lumbar region" (verbatims), but those are not sustainable solutions. Some respondents left comments (104) about what could be done to improve the actual bedpan (a gel cushion or padded support was proposed at 24.0%, a more anatomical shape at 22.0%, or even a more comfortable material at 21.0%) and thus help patients to be more comfortable and feel less pain using a bedpan. Gattinger et al.'s study states, "These findings suggest that innovations in bedpan models are necessary" [9]. Our study confirms their suggestion. Moreover, the aim of this study and the following one with patients directly is to determine the functionalities of a new bedpan, more comfortable and adapted to different morphologies. This will have an impact on the industry through a new design.

In line with the nursing metaparadigm [5, 7], which specifies the person, the environment, health, and care as central, our study shows a non-conformity of pools to meet the fundamental concepts of the nursing metaparadigm. In this transformative action, nursing intervention will take place with the patient, in a comprehensive process. This emphasis highlighted a long-standing shortcoming, first highlighted by interviewing patients on a small scale [9], then caregivers on a larger scale [19].

## Conclusion

The purpose of the current study was to assess the comfort of the patient at bedpan use perceived by the professional and the ease of manipulating the bedpan by the professional. The findings suggest that the pain is surely caused by the pressure from the bedpan on the dorsal decubitus due to the supine position. It also suggests that caregivers perceived the patients' feelings about using the bedpan well. At least they feel concerned about the patient's well-being. Caregivers already try to make the most of this time of intimacy, taking care to allow the patient the necessary time, to provide him with toilet paper and to ensure that his legs are covered by the sheet for greater privacy, in addition to closing the door to his room. When the patient is in a double room, the room-mate is asked to step out for a few moments, or if this is not possible, the curtain separating the beds is drawn. As some caregivers pointed out in the verbatims, alternatives for greater comfort have been found, such as towels positioned under the lumbar region. But this type of installation also requires several trials. Also, they do not seem fully satisfied with the current bedpan. A new bedpan with an enlarged seat-engaging surface and a more comfortable material may be a research avenue. Furthermore, 30.5% responded with an exposure time < 5 min because an extended time can be perceived as unfavorable (caregiver is not responsive to the bell). On the other hand, 11.9% answered a time of more than 11 min and 0.5% more than 15 min. These longer delays may be a sign of forgetting to remove the bedpan or the consequence of an increased workload for the caregiver. This extension of time spent on the bedpan can induce discomfort and pain and alter the skin condition [20]. For 63.1% of the respondents, incomplete elimination of the patient is never because of the lack of time. It would be interesting to evaluate if the longer the time spent on the bedpan to eliminate is, the more detrimental this is to the patient's skin condition.

## Limitation

The main limitation of this study is to investigate patients' comfort through a questionnaire intended for professionals. However, our results are in line with those of previous studies on this subject. Moreover, we also wanted to target the opinion of professionals on their use of bedpans. One may wonder whether caregivers who answered the questionnaire in the majority are not the majority of those who are the least satisfied with the current bedpan. Since pain and intimacy are subjective concepts, we can't be satisfied with professionnals' feelings only.

To validate our hypotheses and the professionnals' feelings about the patients, we need to investigate them directly. A study is currently underway to gather the opinions of over 300 patients in several hospitals. Relevance for clinical practice.

Two significant clinical consequences are to be highlighted:

1) the importance of considering the physical difficulties of caregivers in the installation and removal, and 2) the impact of patient comfort on elimination and well-being.

These results make it possible to begin a search for a solution to the problem highlighted locally but more generally as an overview in a German-Swiss survey [9] or a British article [8]. They also allow a new study on the subjective but practical aspect of discomfort, pain, and objective skin condition change after using the bedpan, this time with patients from the same hospital. Moreover, it is interesting to develop new innovative solutions to facilitate elimination, comfort, and well-being.

## Supporting information

**S1 Appendix. Questionary "Evaluation of bedpan by caregivers".**
(DOCX)

**S1 Checklist. STROBE statement—Checklist of items that should be included in reports of observational studies.**
(DOCX)

**S1 Data.**
(XLSX)

## Acknowledgments

The authors thank the professionals for their time and for answering the questions. The research was performed as part of the employment of the authors at the Nantes University Hospital.

The authors have checked to make sure that our submission conforms as applicable to the Journal's statistical guidelines *described here*. There is a statistician on the author team and state which author.

## Author Contributions

**Conceptualization:** Pia Secher Cailleteau.

**Data curation:** Pia Secher Cailleteau.

**Formal analysis:** Pia Secher Cailleteau, Cécile Paille.

**Investigation:** Pia Secher Cailleteau, Lucie Cadon.

**Methodology:** Pia Secher Cailleteau, Lucie Cadon, Cécile Paille, Elise Olivier, Thomas Rulleau.

**Software:** Thomas Rulleau.

**Supervision:** Thomas Rulleau.

**Validation:** Pia Secher Cailleteau, Thomas Rulleau.

**Visualization:** Pia Secher Cailleteau.

**Writing – original draft:** Pia Secher Cailleteau, Thomas Rulleau.

**Writing – review & editing:** Pia Secher Cailleteau, Lucie Cadon, Cécile Paille, Elise Olivier, Thomas Rulleau.

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
