## [Decision Letter · Decision Letter 0]

25 Jan 2024

PONE-D-23-28993Assessment of perceived patient comfort and ease of bedpan handling by caregivers, a cross-sectional survey.PLOS ONE

Dear Dr. Rulleau,

Thank you for submitting your manuscript to PLOS ONE. After careful consideration, we feel that it has merit but does not fully meet PLOS ONE’s publication criteria as it currently stands. Therefore, we invite you to submit a revised version of the manuscript that addresses the points raised during the review process.

We look forward to receiving your revised manuscript.

Kind regards,

Stefaan Six, Ph.D.

Academic Editor

PLOS ONE

Journal Requirements:

2. For studies reporting research involving human participants, PLOS ONE requires authors to confirm that this specific study was reviewed and approved by an institutional review board (ethics committee) before the study began. Please provide the specific name of the ethics committee/IRB that approved your study, or explain why you did not seek approval in this case.

4. For studies involving third-party data, we encourage authors to share any data specific to their analyses that they can legally distribute. PLOS recognizes, however, that authors may be using third-party data they do not have the rights to share. When third-party data cannot be publicly shared, authors must provide all information necessary for interested researchers to apply to gain access to the data. (https://journals.plos.org/plosone/s/data-availability#loc-acceptable-data-access-restrictions) 

Additional Editor Comments:

Both reviewers raised some interesting points which the authors need to address, but in particular, the comments raised by reviewer 1 regarding the mismatch in percentages, should be fully clarified for this manuscript to be considered for further evaluation.

Reviewers' comments:

Reviewer's Responses to Questions

**Comments to the Author**

1. Is the manuscript technically sound, and do the data support the conclusions?

Reviewer #1: Partly

Reviewer #2: Yes

2. Has the statistical analysis been performed appropriately and rigorously? 

Reviewer #1: N/A

Reviewer #2: I Don't Know

3. Have the authors made all data underlying the findings in their manuscript fully available?

Reviewer #1: Yes

Reviewer #2: Yes

4. Is the manuscript presented in an intelligible fashion and written in standard English?

Reviewer #1: Yes

Reviewer #2: Yes

5. Review Comments to the Author

Reviewer #1: 1) In your abstract, you write ”63% find the installation of the bedpan rather tough or very difficult. 59% find the removal of the bedpan rather tough or very difficult.” However, this is incongruent with your figure 1 that shows only 16% (2% very difficult and 14% somewhat difficult) for removal and 13% for installation. Your numbers are actually showing 60% “somewhat not difficult” and “not difficult at all”. This will also need to be addressed in your descriptive findings on pg 14

2) Introduction “hygiene care represents a large group of care, also called nursing” sounds awkward. It may sound better “hygiene care represents a large part of nursing care in general…It is carried out..”

3) Pg 14, under “participants” , change “nursery school” to “nursing school”

4) Pg 14 “ handle is not particularly comfortable for 45.5% of respondents’ is not the same as a rating “neither comfortable nor uncomfortable” from your questionnaire. Likert scale is supposed to have a neutral, so this 45% should be reported as a neutral finding, not a negative finding toward the handle, which is your implication.

5) Pg 15, Discussion 1st paragraph change “source” of poor elimination to “contributes to”

6) Pg 15, Discussion 3rd paragraph. The sentence that refers to Gattinger et al. sounds incomplete. What does “the fact that bedpan sliding off” mean?

7) Table 1, change “et” to “and”

8) Figure 1, if “reversal of the bedpan” is referencing the “tip over” question in your survey , you should write “frequency bedpan tips over” (14% often/very often)

Conclusions are sound that nursing personnel feel that pain/discomfort is a major cause of incomplete elimination, but other than redesigning the bedpan, are there other alternatives? Catheters provide pain and discomfort in a different way and devices such as the PureWick also may create a level of discomfort, especially for people who are continent. Can perceived pain/discomfort be mitigated through other physiological or physiological factors such as providing privacy, better positioning, providing words of encouragement from staff that while the bedpan is awkward, it is functionally the best option at this time in their care. I do like the idea of having a cushion though.

Reviewer #2: This is an interesting study and nicely raises the issues around bedpan use. I think it would benefit from a bit more inclusion in the introduction section on the tissue damage to patients from sheering and friction caused by sliding bedpans under the patient when rolling the patient onto the bedpan is not an option. There is inconsistency in the referencing, these should all be numbered rather than named.

I have made a number of small comments to the various sections as listed below which need to be considered along with careful checking of some grammar and typo errors. This article has good potential for a great read for nurses, other healthcare professionals and possibly healthcare industry.

Ethics statement.

The last sentence stating ‘unfortunately we were unable to obtain approval from an ethics committee’ you need to clarify why you were unable to obtain approval form an ethics committee or possible removed as you have stated that it wasn’t required.

Impact section

Will it also have an impact of the bedpan industry as they may need to look at redesign?

Pilot study

You make reference to the questionnaire given to 7 nurses/carers to test it but what were the outcomes, did you need to make any changes to the questionnaire?

Design section

You refer to the ‘number of years of bedpan use’ – this may benefit from rewording as this is the number of years’ experience of the caregiver using bedpans.

Ethics section

Needs to include that ethics was not required and why.

Results section - Bedpan manipulation and caregivers’ appreciation of bedpan handling

A bit more explanation of the ‘bedpan falls over’ is required as I am unsure what this means.

Discussion section

In the first paragraph it states ‘have difficulty handling the bedpan during breaks and removals’ needs rewording or explaining as unclear what this means.

Conclusion

The ‘findings’ sentence needs to be reworded as the pain is surely caused by the pressure from the bedpan on the dorsal decubitus.

Limitations

Most of the information in this section is more aligned with the discussion rather than clear limitations. This section could do with rewriting and moving the current information into the discussion.

6. PLOS authors have the option to publish the peer review history of their article (what does this mean?). If published, this will include your full peer review and any attached files.

Reviewer #1: No

Reviewer #2: No

---

## [Author Response · Author response to Decision Letter 0]

2 Jun 2024

We thank the reviewers for their interest in our work. You will find the answers below:

Reviewer #1: 

1) In your abstract, you write ”63% find the installation of the bedpan rather tough or very difficult. 59% find the removal of the bedpan rather tough or very difficult.” However, this is incongruent with your figure 1 that shows only 16% (2% very difficult and 14% somewhat difficult) for removal and 13% for installation. Your numbers are actually showing 60% “somewhat not difficult” and “not difficult at all”. This will also need to be addressed in your descriptive findings on pg 14

Thank you for your careful review. We made a terrible mistake in translating our results into the Fig 1 that led to your comment. Please accept our apologies. As we wrote in the abstract, "62.6% find the installation of the bedpan rather tough or very difficult. 59.2% find the removal of the bedpan rather tough or very difficult. We have corrected the table.

We have also modified the presentation of the results in the text to clarify approximations due to rounding.

2) Introduction “hygiene care represents a large group of care, also called nursing” sounds awkward. It may sound better “hygiene care represents a large part of nursing care in general…It is carried out..”

3) Pg 14, under “participants” , change “nursery school” to “nursing school”

We would like to thank reviewer 1 for his comments. We have made the changes in the “introduction” and the “participants” sections.

4) Pg 14 “ handle is not particularly comfortable for 45.5% of respondents’ is not the same as a rating “neither comfortable nor uncomfortable” from your questionnaire. Likert scale is supposed to have a neutral, so this 45% should be reported as a neutral finding, not a negative finding toward the handle, which is your implication.

We have made the changes in the “Descriptive findings of the study” section

5) Pg 15, Discussion 1st paragraph change “source” of poor elimination to “contributes to”

We have made the changes in the “Discussion” section

6) Pg 15, Discussion 3rd paragraph. The sentence that refers to Gattinger et al. sounds incomplete. What does “the fact that bedpan sliding off” mean?

We modified the sentence to a better comprehension: “Gattinger et al. bring up in their survey patients’ fear “that everything may go bedside” and the fact that “they slide off the bedpan”.”

7) Table 1, change “et” to “and”

We have made the changes in the table 1

8) Figure 1, if “reversal of the bedpan” is referencing the “tip over” question in your survey , you should write “frequency bedpan tips over” (14% often/very often)

Conclusions are sound that nursing personnel feel that pain/discomfort is a major cause of incomplete elimination, but other than redesigning the bedpan, are there other alternatives? Catheters provide pain and discomfort in a different way and devices such as the PureWick also may create a level of discomfort, especially for people who are continent. Can perceived pain/discomfort be mitigated through other physiological or physiological factors such as providing privacy, better positioning, providing words of encouragement from staff that while the bedpan is awkward, it is functionally the best option at this time in their care. I do like the idea of having a cushion though.

We change as your suggestion “frequency bedpan tips over” and add the sentence “Caregivers already try to make the most of this time of intimacy, taking care to allow the patient the necessary time, to provide him with toilet paper and to ensure that his legs are covered by the sheet for greater privacy, in addition to closing the door to his room. When the patient is in a double room, the room-mate is asked to step outside for a few moments, or if this is not possible, the curtain separating the beds is drawn. As some caregivers pointed out in the verbatims, alternatives for greater comfort have been found, such as towels positioned under the lumbar region. But this type of installation also requires several trials. »

Reviewer #2: 

This is an interesting study and nicely raises the issues around bedpan use. I think it would benefit from a bit more inclusion in the introduction section on the tissue damage to patients from sheering and friction caused by sliding bedpans under the patient when rolling the patient onto the bedpan is not an option. There is inconsistency in the referencing, these should all be numbered rather than named.

We thanks reviewer 2 for this comment. We added some lines about tissue damage “Stage 0 or 1 pressure sores are common when bedpans are used, as the rigid surface of the bedpan impairs blood circulation and thus local tissue oxygenation. When the bedpan is removed, redness may be visible, which does not disappear immediately.1“

I have made a number of small comments to the various sections as listed below which need to be considered along with careful checking of some grammar and typo errors. This article has good potential for a great read for nurses, other healthcare professionals and possibly healthcare industry.

We would like to thank proofreader 2 and hope that the changes he has made meet his expectations. We had this sentence in discussion section: “Moreover, the aim of this study and the following one with patients directly is to determine the functionalities of a new bedpan, more comfortable and adapted to different morphologies. This will have an impact on the industry through a new design. »; and these words in “impact section” : “and on the bedpan industry.”

Ethics statement.

The last sentence stating ‘unfortunately we were unable to obtain approval from an ethics committee’ you need to clarify why you were unable to obtain approval form an ethics committee or possible removed as you have stated that it wasn’t required.

In the specific case of this study-involving healthcare’s, with no changes to our clinical practice and no randomization, French legislation about biomedical research does not require authorization from the competent authorithies and approval from the ethics committee. An explanatory letter is attached from the management of the research department at Nantes University Hospital.Impact section

Will it also have an impact of the bedpan industry as they may need to look at redesign?

We added the sentence « The aim of this study and the following one with patients directly is to determine the functionalities of a new bedpan, more comfortable and adapted to different morphologies. This will have an impact on the industry through a new design. »

Pilot study

You make reference to the questionnaire given to 7 nurses/carers to test it but what were the outcomes, did you need to make any changes to the questionnaire?

We added the sentence “This pilot evaluation validated that the questionnaire had been properly understood, without any misinterpretations.” in the design section.

Design section

You refer to the ‘number of years of bedpan use’ – this may benefit from rewording as this is the number of years’ experience of the caregiver using bedpans.

We modify the sentence “the number of years’ experience of the caregiver using bedpans” in the design section.

Ethics section

Needs to include that ethics was not required and why.

In the specific case of this study-involving healthcare’s, with no changes to our clinical practice and no randomization, French legislation about biomedical research does not require authorization from the competent authorithies and approval from the ethics committee. An explanatory letter is attached from the management of the research department at Nantes University Hospital. We had this sentence to explain : “French law makes no provision for an ethics committee to be involved in this study.”

Results section - Bedpan manipulation and caregivers’ appreciation of bedpan handling

A bit more explanation of the ‘bedpan falls over’ is required as I am unsure what this means.

We add for more explanation the sentence “It means that respondents admit that urines and stools end up in the bed”. 

Discussion section

In the first paragraph it states ‘have difficulty handling the bedpan during breaks and removals’ needs rewording or explaining as unclear what this means.

We add the sentence “They sometimes have to go over it several times before the patient feels stable on it. When removing the bedpan, care must be taken not to tip it over, or risk introducing urines and stools into the bed.”

Conclusion

The ‘findings’ sentence needs to be reworded as the pain is surely caused by the pressure from the bedpan on the dorsal decubitus.

As you suggest we change for “pain is surely caused by the pressure from the bedpan on the dorsal decubitus due to the supine position”.

Limitations

Most of the information in this section is more aligned with the discussion rather than clear limitations. This section could do with rewriting and moving the current information into the discussion.

We change our last paragraph and add the sentence « Since pain and intimacy are subjective concepts, we can't be satisfied with professionnals’ feelings only. To validate our hypotheses and the professionnals’ feelings about the patients, we need to investigate them directly. A study is currently underway to gather the opinions of over 300 patients in several hospitals. »

Once again, we would like to thank the reviewers for their comments, which helped us to improve our work.

---

## [Decision Letter · Decision Letter 1]

25 Jun 2024

Assessment of perceived patient comfort and ease of bedpan handling by caregivers, a cross-sectional survey.

PONE-D-23-28993R1

Dear Dr. Rulleau,

We’re pleased to inform you that your manuscript has been judged scientifically suitable for publication and will be formally accepted for publication once it meets all outstanding technical requirements.

Kind regards,

Stefaan Six, Ph.D.

Academic Editor

PLOS ONE

Additional Editor Comments (optional):

Reviewers' comments:

Reviewer's Responses to Questions

**Comments to the Author**

1. If the authors have adequately addressed your comments raised in a previous round of review and you feel that this manuscript is now acceptable for publication, you may indicate that here to bypass the “Comments to the Author” section, enter your conflict of interest statement in the “Confidential to Editor” section, and submit your "Accept" recommendation.

Reviewer #1: All comments have been addressed

2. Is the manuscript technically sound, and do the data support the conclusions?

Reviewer #1: Yes

3. Has the statistical analysis been performed appropriately and rigorously? 

Reviewer #1: Yes

4. Have the authors made all data underlying the findings in their manuscript fully available?

Reviewer #1: Yes

5. Is the manuscript presented in an intelligible fashion and written in standard English?

Reviewer #1: Yes

6. Review Comments to the Author

Reviewer #1: Thank you for your hard work to put this together. I, too, would like to see a more comfortable and user friendly bedpan design created

7. PLOS authors have the option to publish the peer review history of their article (what does this mean?). If published, this will include your full peer review and any attached files.

Reviewer #1: **Yes: **Sharon K McLain, BSN, RNC-OB, C-EFM

---

## [Editor Report · Acceptance letter]

2 Jul 2024

PONE-D-23-28993R1 

PLOS ONE

Dear Dr. Rulleau, 

I'm pleased to inform you that your manuscript has been deemed suitable for publication in PLOS ONE. Congratulations! Your manuscript is now being handed over to our production team.

Kind regards, 

on behalf of

Dr. Stefaan Six 

Academic Editor

PLOS ONE